# Hyaluron-Based Bionanocomposites of Silver Nanoparticles with Graphene Oxide as Effective Growth Inhibitors of Wound-Derived Bacteria

**DOI:** 10.3390/ijms25136854

**Published:** 2024-06-22

**Authors:** Anna Lenart-Boroń, Klaudia Stankiewicz, Kinga Dworak, Klaudia Bulanda, Natalia Czernecka, Anna Ratajewicz, Karen Khachatryan, Gohar Khachatryan

**Affiliations:** 1Department of Microbiology and Biomonitoring, Faculty of Agriculture and Economics, University of Agriculture in Kraków, Adam Mickiewicz Ave. 24/28, 30-059 Kraków, Poland; klaudia.kulik@student.urk.edu.pl; 2Diagnostyka S.A. Medical Microbiological Laboratory, Na Skarpie 66, 31-913 Kraków, Poland; 3Department of Forest Ecosystems Protection, Faculty of Forestry, University of Agriculture in Kraków, 29 Listopada Ave. 46, 31-425 Kraków, Poland; 4Scientific Circle of Biotechnologists, Faculty of Biotechnology and Horticulture, University of Agriculture in Kraków, 29 Listopada Ave. 54, 31-425 Kraków, Poland; 5Laboratory of Nanomaterials and Nanotechnology, Faculty of Food Technology, University of Agriculture in Kraków, Balicka Street 122, 30-149 Kraków, Poland; karen.khachatryan@urk.edu.pl; 6Department of Food Quality Analysis and Assessment, Faculty of Food Technology, University of Agriculture in Kraków, Balicka Street 122, 30-149 Kraków, Poland; gohar.khachatryan@urk.edu.pl

**Keywords:** antibacterial activity, graphene oxide, hyaluronic acid, silver nanoparticles, topical dressings, wound infections

## Abstract

Keeping wounds clean in small animals is a big challenge, which is why they often become infected, creating a risk of transmission to animal owners. Therefore, it is crucial to search for new biocompatible materials that have the potential to be used in smart wound dressings with both wound healing and bacteriostatic properties to prevent infection. In our previous work, we obtained innovative hyaluronate matrix-based bionanocomposites containing nanosilver and nanosilver/graphene oxide (Hyal/Ag and Hyal/Ag/GO). This study aimed to thoroughly examine the bacteriostatic properties of foils containing the previously developed bionanocomposites. The bacteriostatic activity was assessed in vitro on 88 Gram-positive (n = 51) and Gram-negative (n = 37) bacteria isolated from wounds of small animals and whose antimicrobial resistance patterns and resistance mechanisms were examined in an earlier study. Here, 69.32% of bacterial growth was inhibited by Hyal/Ag and 81.82% by Hyal/Ag/GO. The bionanocomposites appeared more effective against Gram-negative bacteria (growth inhibition of 75.68% and 89.19% by Hyal/Ag and Hyal/Ag/Go, respectively). The effectiveness of Hyal/Ag/GO against Gram-positive bacteria was also high (inhibition of 80.39% of strains), while Hyal/Ag inhibited the growth of 64.71% of Gram-positive bacteria. The effectiveness of Hyal/Ag and Hyal/Ag/Go varied depending on bacterial genus and species. *Proteus* (Gram-negative) and *Enterococcus* (Gram-positive) appeared to be the least susceptible to the bionanocomposites. Hyal/Ag most effectively inhibited the growth of non-pathogenic Gram-positive *Sporosarcina luteola* and Gram-negative *Acinetobacter*. Hyal/Ag/GO was most effective against Gram-positive *Streptococcus* and Gram-negative *Moraxella osloensis*. The Hyal/Ag/GO bionanocomposites proved to be very promising new antibacterial, biocompatible materials that could be used in the production of bioactive wound dressings.

## 1. Introduction

Keeping skin wounds clean in small animals is a big challenge, which is why they often become infected. This creates a risk of infection, which results in prolonged or impaired treatment, thus increasing the final treatment cost [1]. In veterinary medicine, the treatment of wound infections is based on broad-spectrum antibiotic administration, which—if overused, unnecessarily used, or used for too long—contributes to the emergence of antimicrobial resistance in bacteria [2]. Most bacterial pathogens found in small animals are also commonly found in humans, which is a potential threat of infection for anyone in contact with a sick animal. Among commonly listed bacteria of zoonotic potential, *Staphylococcus aureus*, *S. pseudinternedius*, *E. coli*, and *Salmonella*, including multidrug-resistant variants of these bacteria, are most commonly listed [3,4,5].

Nowadays, contamination and infections are the most challenging problems in wound care. Commonly used dressings like gauze, bandages, and low-adhesion dressings are usually not efficient enough in terms of preventing bacterial infections [6]. Wound infections can have very severe results, including chronic wound states and tissue necrosis, and can even lead to limb amputation or death. As already shown by [1] or [2] and comprehensively examined in our earlier study [7], the wounds of companion animals can act as habitats for a wide variety of bacterial pathogens, including human pathogens, such as *Proteus mirabilis*, *Staphylococcus aureus*, *Pseudomonas aeruginosa*, *Klebsiella* spp., *Enterococcus faecalis*, *E. faecium*, and *E. coli*. To overcome the problem of wound infections, including particularly problematic infections with antibiotic-resistant bacteria, research has intensified the development of new biocompatible materials with antimicrobial properties that have potential applications in the production of wound dressings [6,8,9,10,11,12,13].

The ideal materials used to provide wound protection or support healing processes should enhance tissue regeneration, such as natural polymer-based hydrogels [14]; they should prevent bacterial biofilm formation or even have antimicrobial properties to prevent the growth of potential bacterial pathogens [9,10,11,12,13]. Among the natural polymers that are promising agents with wound-healing properties, hyaluronic acid is one of the most widely examined. This is due to its ubiquity in vertebrates as a membrane-forming polymer, its involvement in various biological processes, such as cell differentiation or embryonic development, leukocyte chemotaxis, and immunomodulating mediator expression promotion, which is highly beneficial in terms of wound healing [14,15]. Hyaluronic acid also shows anti-angiogenic and anti-inflammatory properties; it allows for tissue irrigation and provides osmotic balance, which contributes to skin hydration and decreases transepidermal water loss [14,16].

In recent years, a variety of metal ion and metal nanoparticle-based materials have been developed that have promising antibacterial properties. These include the biohybrid-based targeted delivery of antibacterial agents using a variety of nanocarriers [11,12] and the application of transition metal nanomaterials and their composites for photodynamic, photothermal, and ion release antibacterial activity [13]. Silver nanoparticles are among the most effective non-antibiotic antibacterial agents against Gram-positive and Gram-negative bacteria as well as fungi and viruses [17,18]. Research also shows that nanoAg can have anti-inflammatory effects and contribute to accelerating wound healing [17,19]. Finally, graphene oxide has also proven to have antibacterial activity, and graphene-formed platforms can be functionalized via the addition of, e.g., metal nanoparticles [9]. It is supposed that GO enhances the surface functionality, surface-to-volume ratio, and antimicrobial properties of metal nanoparticles. Graphene-based materials and hydrogels can interact with bacterial cells, preventing biofilm formation by, e.g., the entrapment of bacteria and stacking the cells, as well as via the destruction of extracellular polymeric substance surrounding a biofilm by the sharp GO edges [9]. For this reason, graphene can also be considered a valid non-antibiotic compound that is effective in preventing wound infections and improving wound healing.

Encouraged by the preliminary results of experiments conducted on the bionanocomposites of hyaluronic acid with nanosilver (Hyal/Ag) and hyaluronic acid with nanosilver and graphene oxide (Hyal/Ag/GO), which showed their bacteriostatic properties against environmental strains of *E. coli*, *Staphylococcus* spp. and *Bacillus* spp., we decided to go further. This study aimed to examine the bacteriostatic properties of the previously formed and characterized bionanocomposites of nanosilver and nanosilver with graphene oxide in a hyaluronate matrix (Hyal/Ag and Hyal/Ag/GO) against Gram-positive and Gram-negative bacterial pathogens, opportunistic pathogens, and non-pathogenic bacteria isolated from the wounds of small animals.

The ultimate goal of this study was to obtain reliable data concerning the bacteriostatic activity of Hyal/Ag and Hyal/Ag/GO bionanocomposites. The experiment was based on a large number of Gram-positive and Gram-negative bacteria of clinical origin and a comparison of the obtained results with those previously observed concerning environmental bacteria.

## 2. Results

In order to ascertain the structure and morphology of the composites obtained, images were captured using scanning electron microscopy (SEM) (Figure 1A,B). The results of the SEM analysis indicate the presence of a vesicular surface structure in the composites. The presence of numerous bright spots suggests the presence of silver nanoparticles within the vesicles. To substantiate the preceding hypothesis, further images were captured utilizing the TEM detector (Figure 1C,D), thereby corroborating the initial conjecture. In both composites (Hyal/Ag and Hyal/Ag/GO), numerous clusters of silver nanoparticles with sizes ranging from a few to several nanometers were observed.

The bacteriostatic activity of the obtained bionanocomposites was examined on 88 (51 Gram-positive and 37 Gram-negative) bacterial strains isolated from the wounds of companion animals. Overall, the growth of 61 (69.32%) was inhibited via the application of the Hyal/Ag bionanocomposite and the growth of 72 (81.82%) was inhibited by Hyal/Ag/GO (Table 1). There were differences between the effect on Gram-positive and Gram-negative bacteria, with a visibly stronger effect on Gram-negative (growth inhibition of 75.68% strains vs. 67.71% for Hyal/Ag and 89.19% vs. 80.39% for Hyal/Ag/GO, respectively) than Gram-positive bacteria (Table 1). There were also clear differences in the effectiveness of the bionanocomposites against different species and groups of bacteria. Both Hyal/Ag and Hyal/Ag/GO inhibited the growth of all *Streptococcus* spp. and non-pathogenic Gram-positive bacteria isolated from animal wounds. Table 1 also compares the results obtained in this study concerning the growth inhibition observed in the initial study, i.e., on environmental strains of bacteria [18], the growth of which was also inhibited (100% of Gram-positive strains for Hyal/Ag and Hyal/Ag/GO; 100% of Gram-negative for Hyal/Ag and 91.67% for Hyal/Ag/GO). Of the Gram-positive strains, *Enterococcus* spp. appears the least susceptible to the Hyal/Ag and Hyal/Ag/GO bionanocomposites (33.33% and 41.67% of inhibited strains, respectively). Within the group of wound-derived Gram-negative bacteria, Hyal/Ag inhibited the growth of 100% of *Acinetobacter* spp. and 92.31% of opportunistic pathogens, whereas both Hyal/Ag and Hyal/Ag/GO were the least effective against *Proteus* spp. (none inhibited by Hyal/Ag and 33.33% by Hyal/Ag/GO, Table 1).

The effectiveness of the bacteriostatic effect of the bionanocomposites against bacteria was also interpreted in terms of the size of the growth inhibition zones on and around the applied Hyal/Ag and Hyal/Ag/GO foils (Figure 2, Figure 3 and Figure 4). Figure 2 shows the differences between the bacteriostatic activity of the applied bionanocomposite foils against Gram-positive and Gram-negative bacteria. As can be seen, Hyal/Ag/GO proved more effective against both groups than Hyal/Ag, and the growth inhibition of Gram-negative bacteria was higher than Gram-positives (mean growth inhibition zones of 13.1 and 14.0 mm caused by Ag and Ag/GO for Gram-negatives while 11.6 and 13.8 mm for Gram-positives). As can be seen in Figure 3, the smallest growth inhibition was observed for Gram-positive *Enterococcus* (6.08 and 6.04 mm for Hyal/Ag and Hyal/Ag/GO, respectively) and Gram-negative *Proteus* (0, i.e., no inhibition and 6.33 for Hyal/Ag and Hyal/Ag/GO, respectively). On the other hand, the highest mean growth inhibition zones were caused by Hyal/Ag against non-pathogenic Gram-positive isolates and Gram-negative *Acinetobacter* (19.38 and 17.80 mm, respectively). Hyal/Ag/GO was the most effective against non-pathogenic Gram-positives, Gram-positive Streptococcus, and opportunist Gram-negative pathogens (19.38, 17.63, and 16.31 mm, respectively). Among the examined groups of bacteria, Gram-negative *Proteus* and Gram-positive *Enterococcus* and *S. aureus*, as well as non-pathogenic Gram-positive bacteria, differ significantly (*p* < 0.05) in their reaction relating to the examined bionanocomposites (Appendix A). The varying reactions of the bacterial strains to the applied bionanocomposite foils are also documented in Figure 4.

The individual growth inhibition zones varied largely not only between the groups of bacteria but also within these groups (Appendix A). For instance, the largest growth inhibition zone observed for Hyal/Ag was 25 mm, which was recorded for *Sporosarcina luteola*. It was followed by 24.5 mm for *Staphylococcus devriesei*, *S. saprophyticus*, and *S. lentus*. On the other hand, no inhibition was also observed for a number of *Staphylococcus* spp. (i.e., *S. schleiferi*, *S. pseudintermedius*, *S. aureus*, *S.pasteuri*, and *S felis*). For Hyal/Ag/GO, the three largest growth inhibition zones were as follows: 25 mm for *S. capitis*, 23 mm for *E. coli*, and 21.5 mm for *Microbacterium maritypicum*, *S. pseudintermedius*, *Macrococcus canis*, and again, *E. coli*. No inhibition was observed for 16 strains: *Acinetobacter pittii*, *Enterococcus faecium*, *E. faecalis*, *E. coli*, *S. pseudintermedius*, *S. pasteuri*, *P. aeruginosa*, *Proteus mirabilis*, *P. vulgaris*, and *Enterobacter hormaechei*. Thus, within the species of *Staphylococcus pseudintermedius* and *E.coli*, there were both the most and least susceptible strains.

Within the group of Gram-positive bacteria, there were three *Staphylococcus* spp. resistant to both Hyal/Ag and Hyal/Ag/GO bionanocomposities Two strains of *S. pseudintermedius* (one was characterized by MSb (macrolide and streptogramin b-type resistance) and was methicillin-resistant; the second one was characterized by constitutive MLSb resistance (Appendix A [7]). The third was *S. pasteuri*, which was methicillin-resistant and also resistant to all seven antibiotics examined in the study [7]). Out of Gram-positives that were resistant to both Hyal/Ag and Hyal/Ag/GO, there were also six. *E. faecalis* and one *E. faecium*, all without multiple resistances or specific resistance mechanisms.

Importantly, both Hyal/Ag and Hyal/Ag/GO composites inhibited the growth of five multidrug-resistant *S. pseudintermedius* strains with inducible and constitutive types of resistance to MLSb antibiotics (Appendix A). One multidrug-resistant (MDR) MRSA was inhibited by Hyal/Ag/GO only. In the case of Gram-negative bacteria, the bionanocomposites proved effective against multidrug-resistant *Aeromonas veronii*, two ESBL-positive, multidrug-resistant (MDR) *E. coli*, ESBL-negative MDR *E. coli*, *Brevundimonas diminuta*, and *Pseudomonas putida*. One MDR *Proteus mirabilis* was inhibited by Hyal/Ag/GO only.

Within the group of Gram-negative bacteria, there were five strains resistant to both bionanocomposites, *E. coli* (resistant to six antimicrobials), *Enterobacter hormaechei* (resistant to five antimicrobials), *Proteus mirabilis* (resistant to nine antimicrobials and ESBL-positive), *P. vulgaris* (resistant to three antimicrobials), and *Pseudomonas aeruginosa* (resistant to four antimicrobials). There was no statistically significant correlation observed between the number of antibiotics that the bacteria were resistant to and their reaction (i.e., growth inhibition zone) to the bionanocomposites.

Statistical analysis of the results showed that the differences in the growth inhibition between the Hyal/Ag and Hyal/Ag/GO, as well as between Gram-positive and Gram-negative bacteria, were statistically not significant (*p* < 0.05). This is probably due to the very distinct differences in the effectiveness of the bionanocomposites against individual groups of bacteria (which were, on the other hand, statistically significant, *p* < 0.05; Appendix A).

## 3. Discussion

The effective prevention and treatment of bacterial infections is one of the most important factors in wound healing. Normal wound healing is a complex and ordered biological process composed of the following four stages: hemostasis, inflammation, proliferation, and remodeling [20]. Importantly, wound infections caused by previous colonization by bacteria or other microorganisms are a major challenge in terms of wound management as they induce severe inflammatory reactions that delay and impair wound healing [21]. Even though bacteria are typical components of animal and human skin microbiota, and even wounds, bacterial biofilm formation is considered a critical threshold that may start the severe process of wound infection that may lead to significant mortality and morbidity [21]. In human patients, methicillin-susceptible and methicillin-resistant *S. aureus*, *E. coli*, *Klebsiella pneumoniae*, and *Pseudomonas aeruginosa* are the prevailing bacterial species involved in wound infections [21,22]. In animals, *Staphylococcus pseudintermedius*, *E. coli*, and *Enterococcus faecalis* are the prevailing pathogens found in wounds [7]. Those above are listed among critical and high-priority pathogens listed by the World Health Organization [23].

Conventional wound dressings (made of cotton gauze) are applied to shield wounds from the external environment, but recent needs to enhance therapeutic actions to improve wound treatment have led to extensive research on enhanced and functionalized dressings. Such wound dressings could contain therapeutic complexes that improve and accelerate wound healing [24]. One of the very important functions of bioactive wound dressings is to prevent bacterial infection. Once integrated into dressings, antimicrobial agents can act effectively and directly against bacteria. Traditional antibiotics, even though effective in vitro against wound-infecting bacterial pathogens, have become rather a risky choice in wound infection treatment. Due to their misuse and abuse, antimicrobial resistance has spread rapidly among bacteria, including those listed as being the most prevalent in wound infections.

For this reason, in this study, we aimed to assess the bacteriostatic effectiveness of two bionanocomposites of nanosilver and nanosilver with graphene oxide in a hyaluronic acid matrix [18], which—in the case of promising results—could serve as the basis for bioactive wound dressings. Detailed chemical and physical characteristics of the obtained composites, i.e., the molecular weight, size and shape, FTIR spectra, UV-Vis absorption spectra, thickness, and the mechanical properties of the composites, as well as contact angle determination, are provided in a previous study [18].

The bacteriostatic effect of the bionanocomposites was examined against 88 bacterial strains isolated from the wounds of small animals and characterized in a previous study [7]. These bacteria comprised 51 Gram-positive and 37 Gram-negative isolates, the growth inhibition of which varied depending on the type of bionanocomposite used, the group of bacteria (Gram-positive vs. Gram-negative), and their taxonomic position. The Hyal/Ag/GO composite appeared to be more effective than the Hyal/Ag composite against both Gram-positive and Gram-negative bacteria, as shown by the percentage of strains where growth was inhibited and by the growth inhibition zones caused by the application of the foils. It is worth noting that both the Hyal/Ag and Hyal/Ag/GO composites inhibited the growth of eleven multidrug-resistant bacteria, and two multidrug-resistant bacteria were inhibited by the sole application of Hyal/Ag/GO. These observations indicate that combining nanosilver with graphene oxide enhances the effectiveness of the obtained composites when acting as bacteriostatic agents. Similar observations were presented by [25,26]. Angulo-Pineda et al. [26] and Prasad et al. [25] explain this phenomenon by the fact that, on the one hand, AgNPs release Ag^+^ ions that bind to the functional groups of bacterial proteins, leading to denaturation and a loss of DNA replication ability, causing oxidative stress, damaging cell walls and membranes, and increasing their permeability, resulting in cell death [26,27]. Mutalik et al. [13] describe the antibacterial mechanisms of metal ions and nanoparticles in detail. Therefore, the level of bacterial hydrophobicity that allows for biofilm formation decreases when they are affected by AgNPs; AgNPs bind to cytoskeletal proteins (tubulin and actin), which inhibits bacterial proliferation; AgNPs strongly interact with transport proteins (such as galactose-transporting dihydrolipoamide dehydrogenases, ATP-binding cassette transporter, and permease transporter), which prevents bacteria from performing material transport; and they also change cell surface charges, thus modifying permeability, which further results in internal component leakage and cell death. Nanosilver also inhibits the respiration processes of bacteria by, e.g., reducing lactate dehydrogenase activity, which is associated with bacterial respiration. This is due to the replacement of hydrogen atoms from the thiol groups of cysteine by positively charged nanosilver, thus preventing the action of respiratory dehydrogenases [13]. The antibacterial activity of GO, on the other hand, mostly requires direct contact with bacterial cells to damage their membrane by the sharp edges, resulting in morphological changes in the cell structure, the leakage of intracellular components, and changes in transmembrane potential. Larger sheets of GO can cover bacterial cells, preventing their proliferation [28]. Mohammed et al. [29] list the following antibacterial mechanisms of graphene biomaterials: direct contact mechanisms involve interference with bacterial lipids, proteins, and nucleic acids through electrostatic adsorption, hydrogen bonding, and π-π stacking, inducing lipid extraction and protein disruption and nanoknives that inhibit growth and kill bacterial cells. The above-listed interactions also interact with DNA and RNA, which alter their structure and properties. Graphene strongly interacts with cellular proteins due to their π-conjugated structures and an abundance of oxygen-containing groups. Proteins absorb the graphene structures, which leads to their immediate denaturation. Further, after graphene contacts bacterial cells, a hydrophobic interaction called nanoscale dewetting occurs between graphene and the phospholipid molecules of the cell membrane, leading to cell membrane collapse. In general, there are three principal mechanisms of graphene antibacterial actions: the sharp edges of nanoknives that cause cell membrane stress, oxidative stress, with and without reactive oxygen species production, and the wrapping or trapping cell membrane, which blocks membrane transport [29]. The listed properties combined result in the robust and enhanced antibacterial efficacy of the bionanocomposites.

The varying bacteriostatic effectiveness of the Hyal/Ag and Hyal/Ag/GO composites against Gram-positive and Gram-negative bacteria has been observed not only in our study but also in our initial experiments on Hyal/Ag and Hyal/Ag/GO composites that involved environmental strains of *E. coli* (Gram-negative), *Bacillus* spp. (Gram-positive), and *Staphylococcus* spp. (Gram-positive) [18], as well as by [30] and reviewed by, among others, [31]. The lower susceptibility to nanoAg and nanoAg/GO composites of Gram-positive bacteria than Gram-negative bacteria is attributed to the greater thickness of the peptidoglycan layer of Gram-positive bacteria; the negative charge of the peptidoglycan layer inactivates Ag^+^ ions [31]. Interestingly, in our former study, the Gram-negative environmental strains of *E. coli* appeared less susceptible to both Hyal/Ag and Hyal/Ag/GO composites than the environmental Gram-positive strains of *Staphylococcus* and *Bacillus* [18]. This may be caused by the fact that much more vivid and significant differences in the bacterial reactions to Hyal/Ag and Hyal/Ag/GO were observed not between Gram-positive vs. Gram-negative groups but between individual groups thereof, or even within the groups (i.e., differences between individual strains within the same genus and/or species).

When considering the inter-species differences in the reaction to the bionanocomposites observed in this study, it appears that *Enterococcus* spp. was significantly the least susceptible genus among Gram-positive bacteria, whereas *Proteus* was the least susceptible Gram-negative bacteria (*p* < 0.05). One of the main aspects in terms of these observations is that *Enterococcus* spp. are currently among the most important pathogens behind human and veterinary infections and are considered among the most dangerous, highly virulent, and multidrug-resistant pathogens under the acronym ESKAPE [32]. Bacteria of this genus are not only naturally resistant to a number of antimicrobials (cephalosporins, clindamycin, penicillin, and trimethoprim/sulfamethoxazole), but their genome is also highly plastic and capable of acquiring and accumulating genetic determinants of resistance to antibiotics and other chemotherapeutics [32]. Even more importantly, in the context of our observations, Cui et al. [33] observed and characterized nanosilver-resistant *E. faecalis*, suggesting that resistant strains can be easily established and selected using subinhibitory concentrations of Ag. Among Gram-negative bacteria, the least susceptible genus was *Proteus*, for which the AgNP resistance has been very rarely reported. Saeb et al. [34] isolated a *P. mirabilis* strain that is highly resistant to a variety of nanosilver particles and suggested that the biofilm formation ability exhibited by these bacteria may have assisted in the observed nanosilver resistance. Yet, another aspect to consider in terms of the inter-species differences in the reactions to the Hyal/Ag and Hyal/Ag/GO composites is that the strongest bactericidal effect, understood as the percentage of strains where growth was inhibited, was observed for Gram-positive *Streptococcus* (both Hyal/Ag and Hyal/Ag/GO) and for Gram-negative *Acinetobacter* (Hyal/Ag) and opportunistic Gram-negative pathogens (Hyal/Ag/GO). The largest growth inhibition zone was observed for Gram-positive non-pathogenic strains (both Hyal/Ag and Hyal/Ag/GO) and in the group of Gram-negative bacteria—again for *Acinetobacter* (Hyal/Ag) and opportunistic pathogens (Hyal/Ag/GO). The high bactericidal effectiveness of the bionanocomposites against *Acinetobacter* is a very promising observation. This is because one of its species, *A. baumanii*, is among the ESKAPE group of pathogens that are of particular concern [35]; *Acinetobacter* spp. has been characterized as having had the greatest increase in antimicrobial resistance in recent years, and the number of strains isolated from bloodstream infections has also increased dramatically [36]. Similar concerns relate to the frequency of multidrug resistance reported in the context of this genus [36] and difficulty in terms of eradication due to the ability of bacteria from this genus to survive under harsh conditions [36].

Not only inter- but also intra-species differences in the reaction of bacteria to the Hyal/Ag and Hyal/Ag/GO bionanocomposites were observed, as, e.g., within the Gram-negative *E. coli* and Gram-positive *S. pseudintermedius* species, there were both strains that showed no reaction to the composite application and strains, the growth inhibition of which was significant (e.g., 20 or 21 mm for *E. coli* and *S. pseudintermedius* × Hyal/Ag, respectively). These differences could not have been attributed to the antibiotic resistance of individual strains because among non-susceptible strains, there were both multidrug-resistant and all-susceptible strains; therefore, Ag and Ag/GO resistance does not seem to correlate with antibiotic resistance.

Another promising result of our study is that the developed bionanocomposites of nanosilver and nanosilver with graphene oxide proved effective in inhibiting the growth of 13 multidrug-resistant (MDR) bacteria. Among them were six Gram-positive (including methicillin-resistant staphylococci) and seven Gram-negative strains (including two ESBL-positive *E. coli*). The bacteriostatic effect of nanoAg and nanoAg combined with graphene against multidrug-resistant bacteria was observed not only in our study but also by Prasad et al. [25], as summarized by [9,31].

Treating and eradicating wound infections is often complicated, primarily due to the ability of bacteria to form biofilms [9,24], and in the case of infections with multidrug-resistant bacteria, the issue becomes even more severe [24]. The biofilm formation ability of bacteria, along with possible deep skin layer involvement in infection, is yet another difficulty when treating wound infections with conventional therapeutic approaches [9]. For this reason, there is still a strong need to develop local therapeutic approaches, such as advanced wound dressing materials that could address the above-mentioned issues. Negut et al. [21] provided a list of ideal wound dressing properties, such as biocompatibility, semi-permeability to water and oxygen, tissue renewal process promotion, and cost-effectiveness. Ideally, such dressings could contain antimicrobial agents that would prevent wound infection or even enable local treatment [21]. The bionanocomposites developed by Khachatryan et al. [18] and further examined in this study seem to fulfill most of the listed properties.

Before stepping further toward the development of biocompatible bactericidal wound dressings, the cytotoxicity of the Hyal/Ag and Hyal/Ag/GO bionanocomposites against skin cells should be examined. It needs to be noted that Prasad et al. [25] stated that Ag nanoparticles show low or no toxicity to human cells, while Angulo-Pineda et al. [26] demonstrated that combining AgNPs with GO significantly decreased cytotoxicity without compromising their high antibacterial activity. However, similar examinations should be conducted for the developed bionanocomposites to confirm their low or lack of cytotoxicity.

## 4. Materials and Methods

### 4.1. Bacterial Strains and Culture Conditions

This study was based on a total number of 88 bacterial strains that were isolated from wound swabs of companion animals, as described in our previous study [7,37]. The examined bacteria comprised 51 Gram-positive and 37 Gram-negative strains, the list of which, along with characterization, is provided in Appendix A. Bacteria, first isolated on selective chromogenic media, such as Chromogenic UTI Medium (ThermoFisher Scientific, Oxford, UK), Cetrimide agar (Biomaxima, Lublin, Poland), and Baird-Parker agar (Biomaxima, Lublin, Poland), were identified to the genus or species level using MALDI-TOF (Matrix-assisted laser desorption/ionisation time-of-flight) technology on a Bruker microflex mass spectrometry instrument (Bruker, Billerica, MA, USA) [7].

Antimicrobial susceptibility patterns were examined and described in our previous study [7], and susceptibility to ozonated nanoencapsulated olive oil was examined in [37].

### 4.2. Synthesis and Characterization of Hyaluronic Acid-Based Nanosilver and Graphene Oxide with Nanosilver Bionanocomposites

The control hyaluronic acid foils (Hyal), bionanocomposites of hyaluronic acid with nanoAg (Hyal/Ag) and hyaluronic acid with nanoAg and graphene oxide (Hyal/Ag/GO) were synthesized and prepared for the tests, as described in our previous study [18]. In brief, the control (Hyal) composite was prepared via the continuous stirring of aqueous sodium hyaluronate (0.8 g in 80.0 mL of water) at 30 °C until a homogeneous and transparent gel was obtained. The Hyal/Ag nanocomposites were prepared as follows: aqueous sodium hyaluronate (0.8 g in 69.4 mL of water) was continuously stirred at 30 °C. Subsequently, a AgNO_3_ solution (1.5 g in 100 mL water) (0.6 mL), 4% *w*/*w* aqueous ammonia (2 mL), and 4% *w*/*w* aqueous xylose (8 mL) were added to the reaction mixture. The resulting mixture was then agitated for 30 min at 60 °C with a magnetic stirrer in a water bath. The suspension of nanoAg in a hyaluronan matrix was cooled to room temperature. Finally, the composite Hyal/Ag/GO was prepared: sodium hyaluronate powder (0.8 g) was added to deionized water (65.4 mL) and continuously stirred at 30 °C. Subsequently, the following solutions were added: an aqueous solution of silver nitrate (1.5 g in 100 mL water) (0.6 mL), aqueous ammonia (4% *w*/*w*) (2 mL), and aqueous xylose (4% *w*/*w*) (8 mL). The resulting reaction mixture was agitated for 30 min at 60 °C with a magnetic stirrer in a water bath. Subsequently, the suspension of graphene oxide (GO) (0.1%, 4 mL) was added to the obtained reaction mixture and mixed for a period of two hours, after which the mixture was cooled to room temperature.

All of the resulting gels (Hyal, Hyal/Ag, and Hyal/Ag/GO) were poured into a clean polypropylene surface and dried at 50 °C to obtain constant-weight, dry, flexible, and transparent films. These were then collected and stored in sterile Petri plates until analysis.

### 4.3. Scanning and Transmission Electron Microscopy (SEM and TEM)

Scanning electron microscopy (SEM) analysis was conducted using a JEOL JSM-7500F microscope (JEOL, Tokyo, Japan) coupled with an AZtecLiveLite Xplore 30 system (Oxford Instruments, Abingdon, UK). Prior to analysis, the samples underwent a coating process involving the deposition of a 20 nm layer of chromium using a K575X Turbo Sputter Coater (Emitech, Ashford, UK). The scanning electron microscope was equipped with a transmission electron microscope (TEM) detector. Samples for transmission electron microscopy (TEM) analysis were prepared by drop coating 10 µL of the sample on carbon-coated 200 mesh copper (100) grids (TAAB Laboratories, Aldermaston, Berkshire, UK).

### 4.4. Bacteriostatic Activity of Nanosilver and Graphene Oxide/Nanosilver Bionanocomposites

To assess the bacteriostatic activity of the prepared bionanocomposites, bacterial strains were transferred to sterile 0.85% NaCl solutions to prepare 0.5 MaCFarland (which represents 1.5 × 10^8^ CFU of bacteria/mL) suspensions, which were streaked on the surface of Mueller–Hinton II agar (Biomaxima, Lublin, Poland). Bionanocomposites, prepared in the form of foils, were UV light-sterilized for 30 min and cut into 5 × 5 mm squares with surface-sterilized scissors. The foil fragments were placed on the surface of bacterial cultures. Foils without the addition of Ag and Ag/GO, i.e., with sole hyaluronic acid, were used as control. The plates were incubated at 36 ± 1 °C for 24 h. After incubation, the results were read by measuring the growth inhibition zones around the foil fragments. Two diameters (the smallest and the largest) were read, and the final result was expressed as a mean of these two reads. All growth inhibition zones were expressed in mm. The larger the growth inhibition zone caused by the application of bionanocomposites, the stronger the bacteriostatic activity.

### 4.5. Statistical Analysis

The analyses were carried out in triplicates, and the results provided in tables and figures are their means. The normality of the results was examined using the Shapiro–Wilk test. The hypothesis concerning the normality of the data was rejected (W = 0.83 and 0.78 for Hyal/Ag and Hyal/Ag/GO, respectively), so non-parametric tests were applied. Kruskal–Wallis tests were used to assess the significance of the differences between (a) the bacteriostatic activity of Hyal/Ag and Hyal/Ag/GO, (b) the bacteriostatic activity of the bionanocomposites against Gram-positive and Gram-negative bacteria, (c) the bacteriostatic activity of the bionanocomposites against bacteria of different species of groups thereof, (d) and the bacteriostatic activity of the bionanocomposites against wound-derived bacteria examined in this study and environmental bacteria, as examined in our earlier research.

## 5. Conclusions

The results indicate that the combination of silver nanoparticles with graphene oxide enhances the antibacterial properties of the former. The incorporation of silver and silver/graphene oxide composites into a hyaluronate matrix may facilitate and accelerate wound healing and regeneration. The bacteriostatic effect of the tested Hyal/Ag and Hyal/Ag/GO bionanocomposites was confirmed in the present study against most wound pathogenic bacteria, including multidrug-resistant clinical strains and ESBL-positive *Enterobacterales*, as well as methicillin-resistant *Staphylococcus* spp. With the increasing importance of companion animals as family members, the need for materials that can effectively prevent and treat wound infections is becoming more pressing. The schematic illustration providing the addressed problem, along with the mechanism of action of the examined bionanocomposites and their potential application as bactericidal wound dressings, is shown in Figure 5.

## Figures and Tables

**Figure 1 ijms-25-06854-f001:**
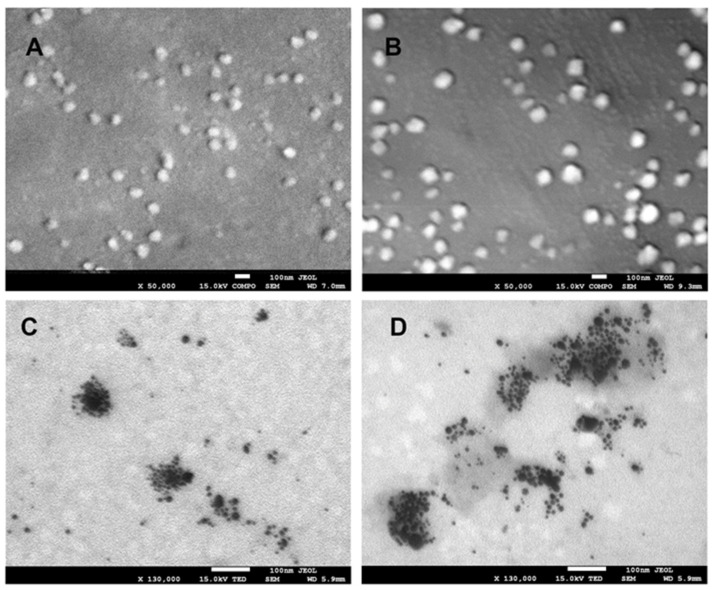
Electron microscopy (SEM) images of Hyal/Ag (**A**) and Hyal/Ag/GO (**B**) at ×50,000 magnification, along with transmission electron microscopy (TEM) detector images of Hyal/Ag (**C**) and Hyal/Ag/GO (**D**) at ×130,000 magnification.

**Figure 2 ijms-25-06854-f002:**
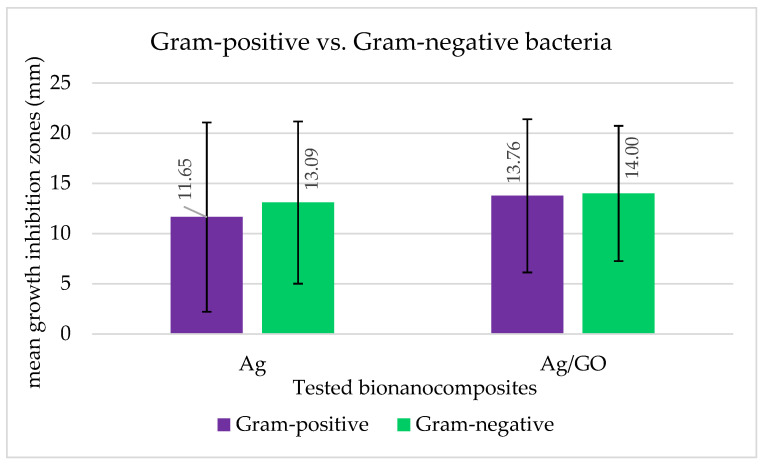
Mean growth inhibition zones (bars represent standard deviations) for Gram-positive and Gram-negative bacteria caused by the application of the tested bionanocmposites of Hyal/Ag and Hyal/Ag/GO. The results are means of three replications (*p* < 0.05). Bars represent standard deviations.

**Figure 3 ijms-25-06854-f003:**
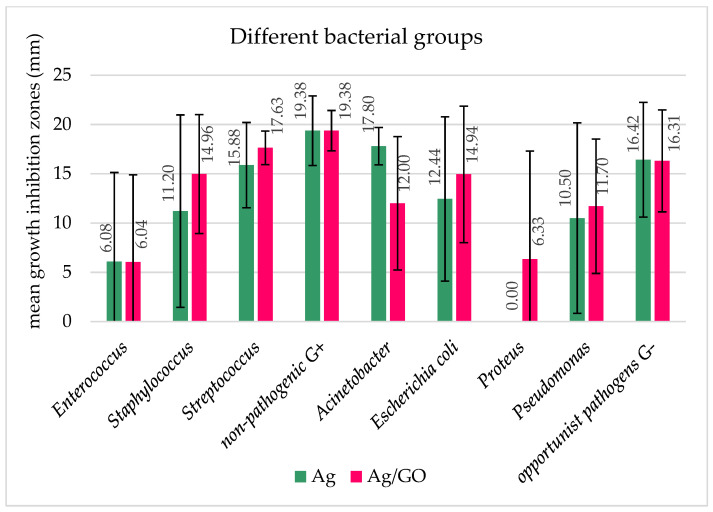
Mean growth inhibition zones (bars represent standard deviations) for different bacterial species and genera of pathogenic and non-pathogenic Gram-positive and Gram-negative bacteria caused by the application of the tested bionanocmposites of Hyal/Ag and Hyal/Ag/GO. The results are means of three replications (*p* < 0.05). Bars represent standard deviations.

**Figure 4 ijms-25-06854-f004:**
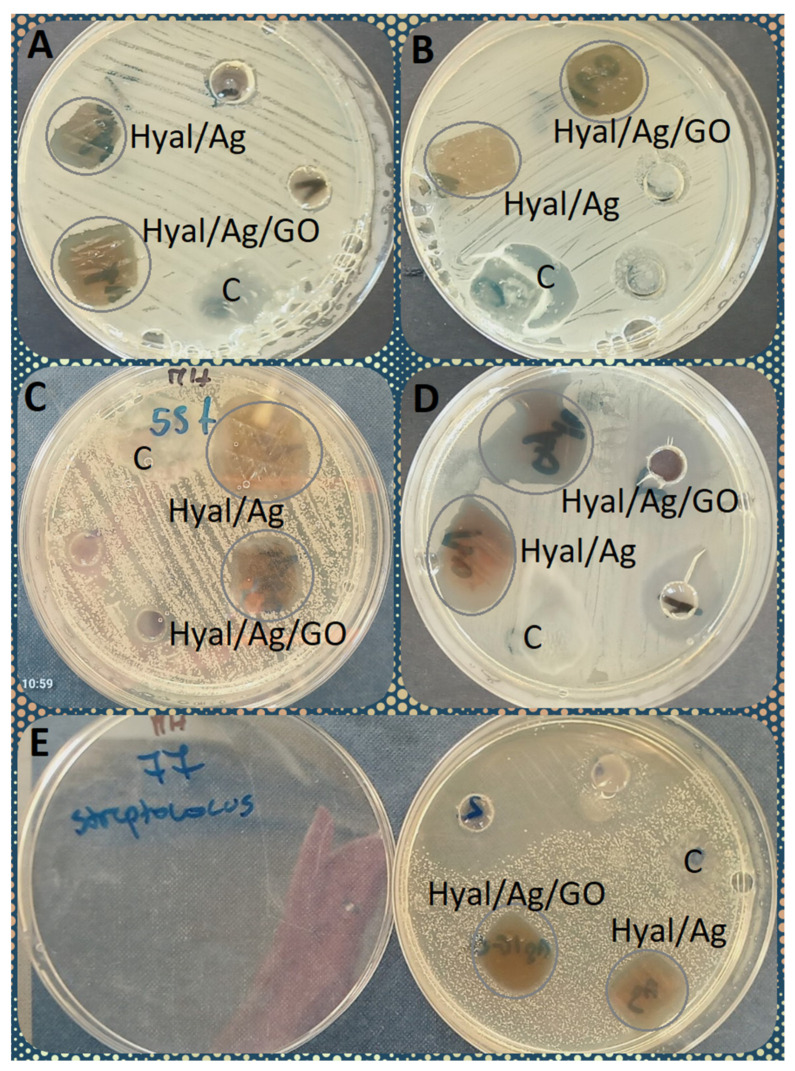
Bacteriostatic activity of the bionanocomposites (Hyal/Ag and Hyal/Ag/GO) compared to the control (C) against the following: (**A**) *Escherichia coli*, (**B**) *Serratia marcescens*, (**C**) *Staphylococcus lentus*, (**D**) *Sporosarcina luteola,* and (**E**) *Streptococcus canis*. The growth inhibition zones are marked with grey circles.

**Figure 5 ijms-25-06854-f005:**
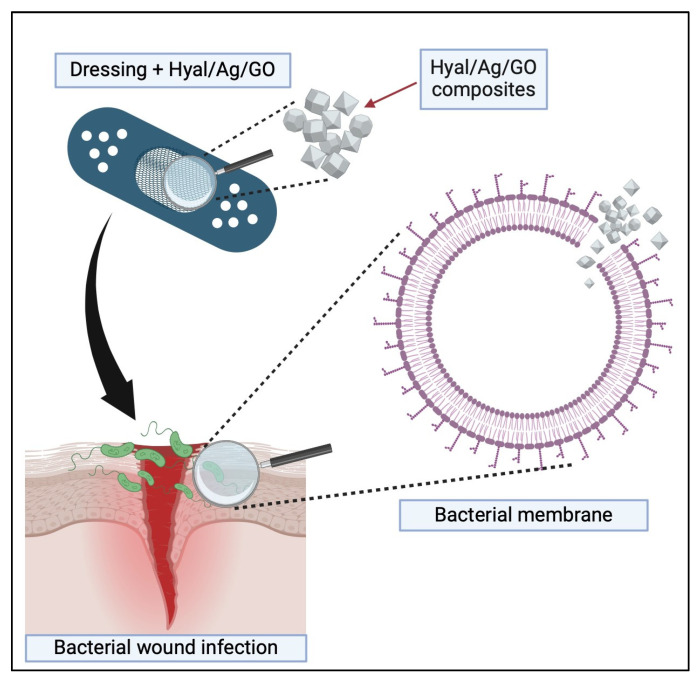
A scheme presenting the potential application of the Hyal/Ag/GO composite examined in this study. Wounds are frequently infected with bacteria that increasingly become antibiotic-resistant. Applying bioactive topical dressings that contain, e.g., the Hyal/Ag/GO composite, the bactericidal effectiveness has been confirmed in this study, allowing the prevention of such infections. This results from the fact that graphene oxide damages bacterial cells, allowing the Ag nanoparticles to act more effectively than without combination with GO.

**Table 1 ijms-25-06854-t001:** Percentage (%) of bacteria of various groups, the growth of which was inhibited by the bionanocomposites.

Species	Tested Composites		
	Hyal/Ag	Hyal/Ag/GO	Control
Gram-positive (n = 51)	67.71	80.39	0
*Enterococcus* (n = 12)	33.33	41.67	0
*Staphylococcus* (n = 27)	62.96	88.89	0
*Streptococcus* (n = 4)	100	100	0
Non-pathogenic G+ (n = 8)	100	100	0
Environmental G+ (n = 5) [18]	100	100	0
Gram-negative (n = 37)	75.68	89.19	0
*Acinetobacter* (n = 5)	100	80	0
*Escherichia coli* (n = 9)	77.78	88.89	0
*Proteus* (n = 3)	0	33.33	0
*Pseudomonas* (n = 5)	60	80	0
Opportunistic G− (n = 13)	92.31	92.31	0
Environmental G− (n = 12) [18]	100	91.67	0
In total (n = 88)	69.32	81.82	0

## Data Availability

The original contributions presented in the study are included in the article/Appendix A, further inquiries can be directed to the corresponding author/s.

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
