# Peer review of "Hyaluron-Based Bionanocomposites of Silver Nanoparticles with Graphene Oxide as Effective Growth Inhibitors of Wound-Derived Bacteria"

_ijms, 2024, doi:10.3390/ijms25136854_

Round 1
Reviewer 1 Report
Comments and Suggestions for Authors
In this manuscript, the authors developed hyaluron-based bionanocomposites of Ag nanoparticles and graphene oxide, and demonstrated its performance for bacteria growth inhibition. This work seems to be useful for this field. However, the following problems should be addressed before further consideration of publication:
1. The title can be revised to be concise where Gram-positive and Gram-negative bacteria don’t need to be distinguished. In the abstract, the innovation can be demonstrated especially the superiority when compared with other researches.
2. All the figures need to be revised with consistent layout and style to improve the readability. The notes in Figure 4 are not clear.
3. A schematic illustration can be created to better show the detailed mechanism, composition, and application property of the samples.
4. In the Introduction section, typical examples of bacterial killing can be briefly introduced. The recent advances should be contained including: 10.1016/j.colsurfa.2024.133295, 10.1002/EXP.20210145.
5. Material characterization of the samples should be added including high-resolution TEM, XPS analysis, and zeta potentials.
6. In Figure 3, the results of “Proteus-Ag” seem to be disappeared. How do you evaluate the results since the error bars are quite large in these types of experiments?
7. The gradient concentration of bacteria in bar plating experiments should be stated.
8. In the manuscript, the depth could be improved if the authors provided some insights of micro-/nano and biological interactions responsible for bacteria growth inhibition of the samples.
Author Response
Dear Reviewer,
the replies are in the attached Word document.
With kind regards,
Anna Lenart-Boroń

Reviewer 2 Report
Comments and Suggestions for Authors
This work have demonstrated bacteriostatic properties of foils containing Hyal/Ag and Hyal/Ag/GO for Gram-positive and Gram-negative bacteria. The bionanocomposites of Hyal/Ag/GO have exhibited superior effect against Gram-negative bacteria and Gram-positive bacteria. The authors have demonstrated that the Hyal/Ag/GO bionanocomposites with potential as antibacterial biocompatible materials. For this manuscript, I have some comments for the authors.
1. For the SEM images of Figure 1, the images were too blurred. This manuscript can be better fi the authors could provide better SEM images.
2. To demonstrate the structure of Hyal/Ag and Hyal/Ag/GO, UV-Vis and FTIR spectra should be provided in the supporting information.
3. To confirm the antibacterial activity of GO, the authors should provide data of the growth inhibition zones. The authors have only provided antibacterial tests of Hyal/Ag and Hyal/Ag/GO.
4. The authors have indicated silver nanoparticles are among the most effective non-antibiotic antibacterial agents. To broaden the introduction, additional references could be cited.
https://doi.org/10.2147/IJN.S392081
Author Response
Dear Reviewer,
the replies are in the attached Word document,
With kind regards,
Anna Lenart-Boroń

Round 2
Reviewer 1 Report
Comments and Suggestions for Authors
All the revisions have been checked.
Reviewer 2 Report
Comments and Suggestions for Authors
The authors have revised the manuscript according to the comments from the reviewers. Furthermore, the authors have provided all figures in a separate .zip file to reveal figures before PDF conversion. With these improvements, the revised manuscript can be published as current version in IJMS.